# Quantum emission from coupled spin pairs in hexagonal boron nitride

Song Li [1,2] ✉, Anton Pershin[1,3] & Adam Gali [1,3,4] ✉

Optically addressable defect qubits in wide band gap materials are favorable candidates for room-temperature quantum information processing. Two-dimensional (2D) hexagonal boron nitride (hBN) is an attractive solid-state platform with great potential for hosting bright quantum emitters and quantum memories, leveraging the advantages of 2D materials for scalable preparation of defect qubits. Although room-temperature bright defect qubits have been recently reported in hBN, their microscopic origin, the nature of the optical transition, and the optically detected magnetic resonance (ODMR) have remained elusive. Here, we connect the variance in the optical spectra, optical lifetimes, and spectral stability of quantum emitters to donor-acceptor pairs (DAPs) in hBN through ab initio calculations. We find that DAPs can exhibit ODMR signals for the acceptor counterpart of the defect pair with an $S = 1/2$ ground state at non-zero magnetic fields, depending on the donor partner and dominantly mediated by the hyperfine interaction. The donor-acceptor pair model and its transition mechanisms provide a recipe for defect qubit identification and performance optimization in hBN for quantum applications.

Isolated optically active atomic defects in wide-band-gap materials serve as single-photon emitters (SPEs), which are key components for quantum information technologies[1,2]. Hexagonal boron nitride (hBN) is a layered van der Waals (vdW) material and a favorable host for SPEs due to its fabrication versatility and compatibility with lithographic processing[3–7]. The observed SPEs in hBN feature high brightness, room-temperature stability, sharp zero-phonon-line (ZPL) peaks around 2 eV, and short excited-state lifetimes[8–16]. Furthermore, coherent control of single electron spins has been recently demonstrated in hBN[17], including systems operating at room temperature[18,19], where electron spin initialization and readout rely on optical excitation and emission of defect spins, a technique known as optically detected magnetic resonance (ODMR).

One major challenge is the identification of the exact defect structures responsible for SPEs and single-spin ODMR centers, which is a prerequisite for realizing deterministic formation and control. The observed photoluminescence (PL) spectra exhibit varying ZPL energies and phonon sidebands (PSBs), and many show similar optical lineshapes[9]. These emissions may originate from various defects, but the similarities in optical lineshape imply the presence of common defect types in diverse crystalline environments[20–22].

An $S = 1/2$ paramagnetic defect with strong hyperfine interaction involving two equivalent nitrogen nuclei has been observed by electron paramagnetic resonance (EPR)[23], and we assigned this signal to the negatively charged $O_N V_B$ defect−i.e., oxygen substituting nitrogen adjacent to a boron vacancy−based on excellent agreement between experimental and simulated EPR spectra[24]. Notably, the existence of the $O_N V_B$ defect was confirmed by subsequent annular dark-field scanning transmission electron microscopy (ADF-STEM) measurements[25]. In addition, carbon and oxygen substitutions were simultaneously observed nearby using the same technique. This provides strong evidence that the extra charge on the $O_N V_B$ defect giving rise to the EPR signal could originate from donor-like substitutions of boron by carbon ($C_B$) or nitrogen by oxygen ($O_N$)[26]. In other words, $C_B$ or $O_N$ may form

[1]HUN-REN Wigner Research Centre for Physics, Budapest, Hungary. [2]Beijing Computational Science Research Center, Beijing, China. [3]Department of Atomic Physics, Institute of Physics, Budapest University of Technology and Economics, Budapest, Hungary. [4]MTA-WFK Lendulet "Momentum" Semiconductor Nanostructures Research Group, Budapest, Hungary. ✉e-mail: li.song@csrc.ac.cn; gali.adam@wigner.hun-ren.hu

donor-acceptor pairs (DAPs) with $O_N V_B$, described as $C_B^+ - O_N V_B^-$ or $O_N^+ - O_N V_B^-$ in the ground state, where the $S = 1/2$ spin state arises from spin density localized around the $O_N V_B^-$ component of the DAP. In this sense, the common defect type is the $O_N V_B$ acceptor, while the variation in optical properties is governed by the type and position of the donor partner.

Here, we perform comprehensive theoretical calculations on the optical properties of DAPs with varying separation distances. We find that the donor ($C_B$ and $O_N$) indeed donates an electron to the $O_N V_B$ defect, rendering it negatively charged. The donor-acceptor distance significantly influences the electronic structure, offering a possible explanation for the ZPL variation observed in experiments. We show that the $O_N - O_N V_B$ DAP is photostable, with quantum yields comparable to those of the isolated negatively charged $O_N V_B$. In contrast, the $C_B - O_N V_B$ pair exhibits metastable dim states at certain separations, which act as non-radiative decay pathways. Spin-flipping within these dim states can mix doublet and quartet multiplets, leading to spin polarization in the $S = 1/2$ ground state of the negatively charged $O_N V_B$ when external magnetic fields lift the Kramers degeneracy. As a consequence, ODMR spectrum[24] may arise from the $C_B - O_N V_B$ defect pair in the $S = 1/2$ ground state when subjected to a constant magnetic field.

## Results and discussion
### Optical properties of carbon and oxygen DAPs
Our modeling relies on recent ADF-STEM results, where single and multiple carbon and oxygen atoms adjacent to the $V_B$ monovacancy—particularly substituting the innermost nitrogen atoms—were imaged[25]. However, monovacancy structures with multiple first-neighbor substitutions do not yield bright ZPL emission at 2 eV, which is frequently observed in experiments, as discussed in Supplementary Note 1. Instead, another interesting phenomenon was

observed in ADF-STEM measurements: carbon and oxygen substitutional defects appear in vacancy-free regions. The presence of $C_B$, $C_N$, and $O_N$ defects can be readily confirmed, while the $O_B$ defect has a high formation energy and is therefore rarely observed[25]. A previous theoretical study indicated the donor characteristics of $C_B$ and $O_N$ defects[26]. We emphasize that the oxygen-related $O_N V_B$ defect may exhibit 2 eV ZPL emission in its negative charge state[24]. Hence, we propose that some of the observed 2 eV SPEs originate from $O_N V_B^-$, with the additional charge supplied by either $C_B$ or $O_N$. In Supplementary Fig. 1, we illustrate that the single-electron levels of $C_B$ or $O_N$ lie higher in energy than the empty $a_1$ state of $O_N V_B$, facilitating electron transfer from the donor to $O_N V_B$. Furthermore, we calculate the isolated defects in different charge states and find that the total energy of the $C_B^+ - O_N V_B^-$ configuration is lower than that of the neutral $C_B^0 - O_N V_B^0$ pair, confirming the stability of the DAP. Notably, charge transfer within DAPs is typically distance-dependent: in wide-band-gap materials with localized donor and acceptor states, the probability of charge transfer rapidly decays as the donor-acceptor distance increases. Therefore, we focus on several DAP structures characterized by relatively short separation distances.

We begin by examining the properties of the $C_B - O_N V_B$ DAPs, as illustrated in Fig. 1. The atomic sites, or coordinates, of the donor species relative to the location of the acceptor are labeled in Fig. 1a. The first-neighbor site, which could form a direct bond to oxygen, was not considered due to a potential repulsive interaction between $O_N$ and $C_B$[12]. The $C_B1$ and $C_B2$ sites are located near the oxygen atom, resulting in a significant influence of the donor on the DAP properties.

The $O_N V_B$ defect possesses $C_{2v}$ symmetry, so we label the in-gap localized defect levels according to their irreducible representations, based on the symmetry operations of the wavefunctions. The corresponding localized wavefunctions are shown in Supplementary Fig. 1.

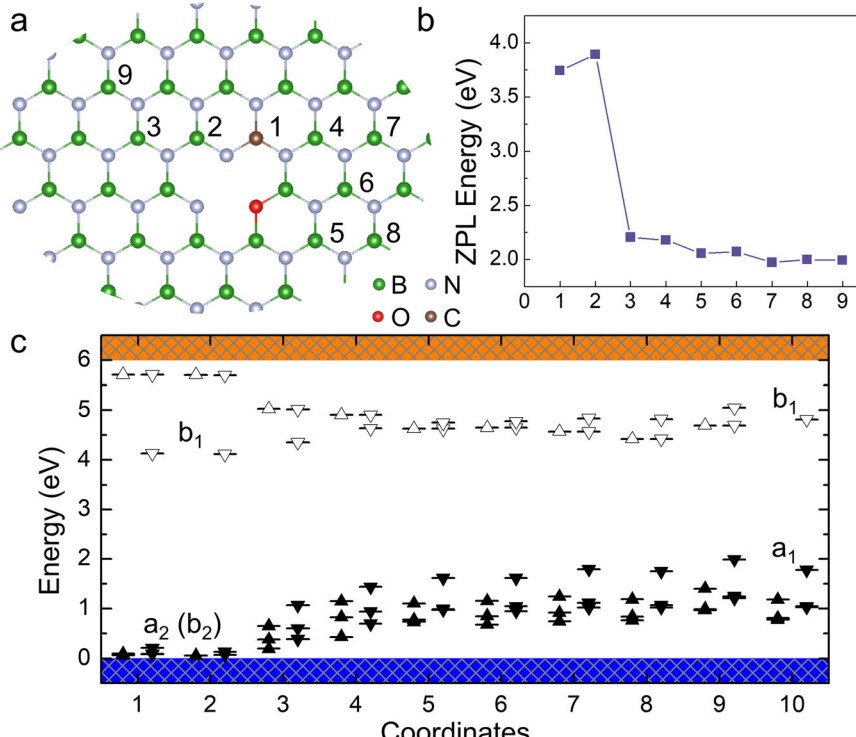

**Fig. 1 | $C_B - O_N V_B$ DAPs in hBN. a** The numbers indicate the positions of $C_B$ at various distances in the hBN lattice (boron and nitrogen atoms are shown as gray and green spheres, respectively). In this panel, $C_B$ is placed at site 1 (brown sphere).
**b** Distance-dependent ZPL energies corresponding to the intrinsically bright optical transition, calculated without including non-radiative processes. Source data are provided as a Source Data file. **c** Electronic structure of the defects in the ground state. Number 10 corresponds to the isolated $O_N V_B^-$ defect, shown for reference. The irreducible representations of defect levels under $C_{2v}$ symmetry are indicated. $a_1$ and $b_1$ denote levels with in-plane localized wavefunctions, while $a_2$ and $b_2$ correspond to out-of-plane extended orbitals. Filled and unfilled triangles represent occupied and unoccupied defect levels within the gap, respectively, and the direction of the triangle indicates spin-majority or spin-minority character.

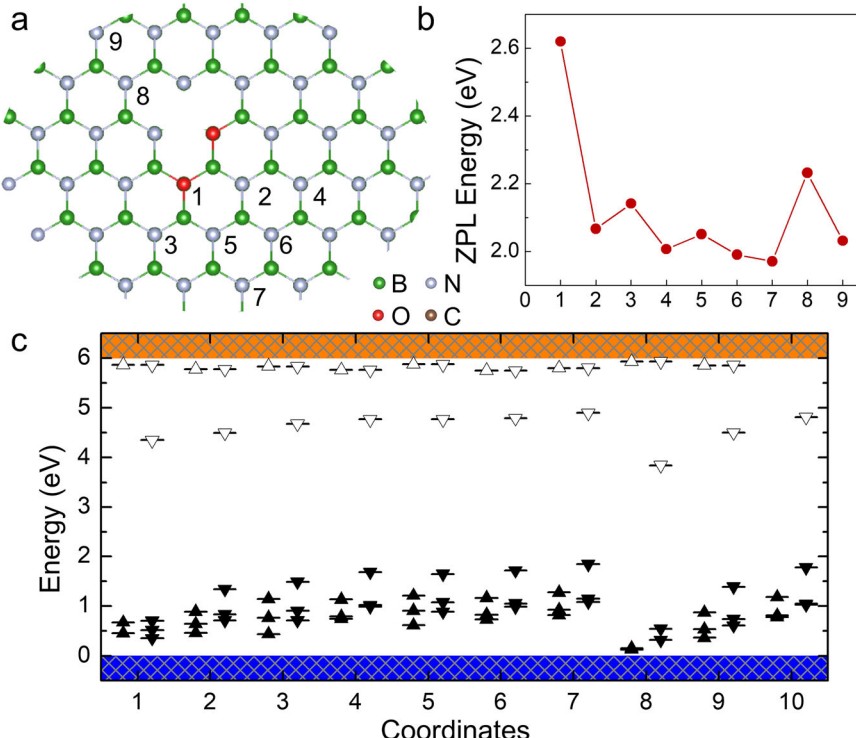

**Fig. 2 | $O_N$–$O_N V_B$ DAPs in hBN. a** The numbers indicate the positions of $O_N$ at various distances in the hBN lattice (boron and nitrogen atoms are shown as gray and green spheres, respectively). In this panel, $O_N$ is placed at site 1 (red sphere). **b** Distance-dependent ZPL energies. Source data are provided as a Source Data file. **c** Electronic structure of the defects in the ground state. Position 10 corresponds to the isolated $O_N V_B^-$ defect, shown for reference. Filled and unfilled triangles represent occupied and unoccupied defect levels within the gap, respectively. The orientation of each triangle indicates spin-majority or spin-minority character.

In the isolated $O_N V_B^-$ defect, the optical transition occurs between the $a_1$ and $b_1$ levels, which are in-plane extended and localized on two equivalent nitrogen atoms along the $C_{2v}$ axis. However, strong interaction at the $C_B 1$ and $C_B 2$ positions alters the ordering of the occupied defect levels: the $a_1$ level shifts downward, while the $a_2$ or $b_2$ levels shift upward. The presence of $C_B$ lowers the symmetry to $C_{1h}$, merging the original $a_2$ and $b_2$ labels. Since the $a_2$ ($b_2$) orbitals are out-of-plane while $b_1$ remains in-plane, the resulting transition dipole moment is oriented out-of-plane with very small magnitude (~0.12 Debye), making the optical transition weak. Thus, the bright emission still originates from the $a_1 \to b_1$ transition. The large energy separation between these levels yields high ZPL energies: 3.7 and 3.9 eV for $C_B 1$ and $C_B 2$, respectively, located in the ultraviolet region.

As the donor–acceptor distance increases, the defect-level ordering reverts to that of the isolated $O_N V_B^-$, and the wavefunction of $C_B$ recovers its $D_{3h}$ symmetry. Another observed trend is that the unoccupied levels of $C_B$ shift downward, while the $b_1$ level shifts upward. In the $C_B 5$ configuration, the empty orbital from $C_B$ lies below the $b_1$ orbital of $O_N V_B$. Nevertheless, the transition dipole moment from $a_1$ to $C_B$ remains small (~0.16 Debye), so we focus on the bright emission localized on $O_N V_B$, where the ZPL energies gradually converge to 1.97 eV. We note that the charge stability of $C_B$–$O_N V_B$ DAPs is broader than that of the isolated $O_N V_B$ defect, as the Coulombic attraction lowers the formation energy and shifts the charge transition levels toward the band edges, as discussed in Supplementary Note 2.

The above-discussed bright emission is a local excitation within $O_N V_B^-$. In addition, the charge transfer process from $O_N V_B^-$ to $C_B^+$ may also result in luminescence, although it is intrinsically dim. The dim metastable state (MS) has two sub-states: an $S = 1/2$ doublet or an $S = 3/2$ quartet, depending on the relative spin orientations within the DAP. With Kohn-Sham density functional theory, we reliably calculate the total energy and geometry of the $S = 3/2$ state. Supplementary Fig. 4 shows the Kohn-Sham energy differences and excitation

energies for which Kohn-Sham defect state pairs can be associated with intrinsically dim and bright optical transitions (without considering non-radiative processes or rates here).

Generally, the bright local excitation is independent of the distance between donor and acceptor, while the dim one shows a strong dependence. The bright emission is the first optical transition in $C_B 3$ and $C_B 4$; however, when $C_B$ resides at larger distances from $O_N V_B$, the dim emission appears at lower energy. This suggests the presence of a non-radiative decay path from the local excited state to the ground state. A simple model to capture the dim emission mechanism can be expressed as

$$E(R_i) = E_{gap} - (E_D + E_A) + \frac{e^2}{\epsilon R_i}, \tag{1}$$

where $E_D$ and $E_A$ are the energy levels of the donor and the acceptor in the band gap, respectively. Here, we use the charge transition levels (CTLs) of $C_B$ $(0| + 1)$ and $O_N V_B(-1|0)$ from a previous study[26]. $\epsilon = 6.93$ is the in-plane dielectric constant of hBN[27], and $R_i$ is the separation distance between donor and acceptor. The last term represents Coulombic interaction, and the reciprocal function leads to a fast convergence of $E(R_i)$ to the CTL difference (1.27 eV) for DAPs with large $R_i$, as shown in Supplementary Fig. 5. The $S = 3/2$ state converges properly and may serve as a reliable reference for predicting the energy of the $S = 1/2$ state, as their energy difference arises from spin-spin exchange interaction, which is typically less than 1 meV. As summarized in Supplementary Fig. 4, the total energy of the $S = 3/2$ state lies 1.68 to 2.36 eV above that of the ground state, depending on DAP separation. This means that the bright excited state (~2.0 eV) and the metastable states may exchange their energy ordering.

$O_N$–$O_N V_B$ DAPs are different from $C_B$–$O_N V_B$ DAPs (see Fig. 2). Even at short distances between $O_N$ and $O_N V_B$, the $O_N V_B^-$ character is still well preserved, likely due to the shallow donor nature of $O_N$. The

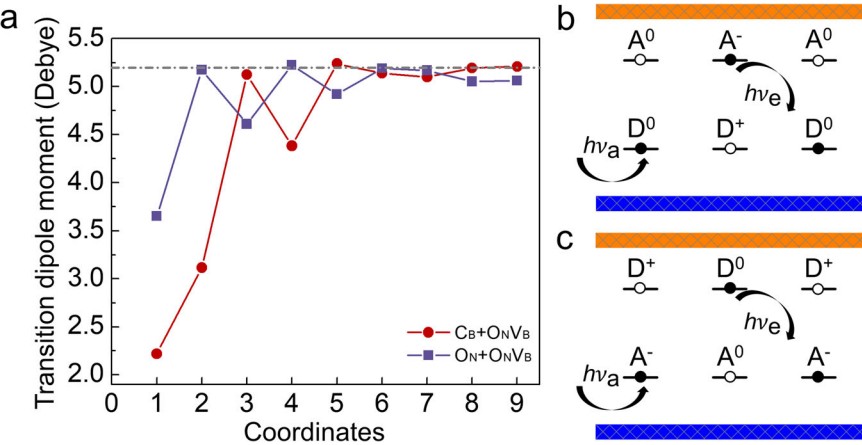

**Fig. 3 | Two types of emission from DAPs. a** Transition dipole moment evolution across the considered defect configurations. The dashed line indicates the transition dipole moment of the isolated $O_NV_B^-$. Coordinates labels are defined in Figs. 1 and 2. Source data are provided as a Source Data file. **b** Type 1 and (**c**) Type 2 mechanisms of optical transitions for DAPs in semiconductors. The two types differ whether charge transfer or direct recombination occurs: the excited electron can be trapped by $A^0$ or $D^+$, altering the charge state before relaxing back to the ground state.

empty levels from $O_N$ lie very close to the conduction band minimum and therefore have negligible influence on the optical transition within $O_NV_B$. Except for the $O_N1$ site, the ZPL energies are all below 2.23 eV. Figure 3a shows the evolution of the transition dipole moment for the bright excitation across the considered configurations, which is significantly stronger than that of the dim transition due to the charge transfer process, as shown in Supplementary Fig. 4. It rapidly converges to the value of the isolated $O_NV_B^-$ as the DAP separation slightly increases. The large energy difference between the charge transition levels of $O_N(0|+1)$ and $O_NV_B(-1|0)$ results in $E(R_i)$ exceeding 2.89 eV; for example, the $S = 3/2$ level of $O_N4$ lies 3.59 eV above the ground state. Accordingly, no metastable states arise for the $O_N^+ - O_NV_B^-$ pair, which makes the optically and spin-active $O_NV_B^-$ defect a photostable emitter, albeit with minor variations in ZPL wavelengths depending on the actual distance between the donor and the acceptor.

**The fluorescence mechanisms in DAP**

The fluorescence mechanisms within DAPs, as shown in Fig. 3b, c, can be classified into two types according to the relative energy levels of the donor and acceptor. If the donor level is lower than the acceptor level, then charge transfer cannot occur spontaneously. Recombination can be induced by illumination from $D^0$ to $A^0$ (neutral charge state) with photon energy $h\nu_a$, creating an ionized donor $D^+$ and acceptor $A^-$. In other words, the electron from the occupied state of $D^0$ is excited to the empty state of $A^0$. The electron returns via radiative decay (charge recombination), and a photon is emitted with energy $h\nu_e$. If the donor level is higher than the acceptor level, the electron from the donor may spontaneously transfer to the acceptor when they are at short distances. Subsequently, photoexcitation converts the charged DAP to its neutral state. The charge transfer from donor to acceptor is distance-dependent, and a recent study investigated this process based on the Marcus theory framework[28]. In contrast to the previous cases, photoexcitation here generally acts on orbitals localized on the acceptor, which is not included in Fig. 3, and may lead to bright emission from the excited state. This is particularly true for the $O_N^+ - O_NV_B^-$ pair. The shallow donor character of $O_N$ places the dim $O_N^0 - O_NV_B^0$ states at significantly higher energies than the bright excited state.

In the $C_B^+ - O_NV_B^-$ DAP, the situation differs, as depicted in the optical loop of Fig. 4a. Optical excitation first drives the doublet ground state to the optically active doublet excited state ($D^+ + A^-$). The small energy difference between the optically active excited state and the intrinsically dim metastable states ($D^0 + A^0$) enables charge transfer from $O_NV_B$ ($A^-$) to $C_B$ ($D^+$) via non-radiative decay. The doublet ($D_s$) and

quartet ($Q_s$) metastable levels are separated by the spin-spin exchange interaction $J$. The ground and optically active doublet excited states are connected to the $D_s$ state by either weak optical transitions or internal conversion (IC), while transitions to the $Q_s$ state occur via intersystem crossing (ISC), mediated by spin-flip processes. Using the same orbitals in both $D_s$ and $Q_s$, it is widely accepted that IC is significantly faster than ISC[29]. Therefore, the $D_s$ state becomes preferentially populated within the metastable manifold. At zero magnetic field, the $D_s$ state provides both radiative and non-radiative decay pathways, shortening the optical lifetime of $O_NV_B^-$ in this DAP compared to that in the $O_N^+ - O_NV_B^-$ DAP. We note that depending on the magnitude of the non-radiative rate ($r1$ rate, see also Supplementary Fig. 6), non-radiative decay to the metastable $D_s$ may dominate, and dim optical transitions may subsequently occur from the $D_s$ DAP state.

At non-zero magnetic fields, the Kramers doublets split in the $C_B^+ - O_NV_B^-$ DAP, where the presence of the $Q_s$ state plays an important role in spin-dependent non-radiative decay. For instance, the higher branch ($m_S = +1/2$) in $D_s$ can mix with the lower branch ($m_S = -3/2$) in $Q_s$. In other words, $D_s$ acquires some $Q_s$ character, and vice versa. Therefore, the $m_S = -1/2$ spin sublevel localized on $O_NV_B^-$ in $D_s$ can become more populated than the $m_S = +1/2$ sublevel through the non-radiative decay channel $r3$ (see Fig. 4). This spin-selective decay populates the $m_S = -1/2$ spin state in the electronic ground state preferentially over the $m_S = +1/2$ state, making the $m_S = -1/2$ level appear brighter. As a consequence, an ODMR signal can be observed for an $S = 1/2$ ground state. The key feature here is the quasi-degeneracy of the $D_s$ and $Q_s$ states in the metastable $C_B^0 - O_NV_B^0$ configuration, which corresponds to a spin pair of $S = 1/2$ (see ref. 30) and $S = 1$ (see ref. 24) defects, respectively. This interaction can lead to spin polarization of the split $S = 1/2$ Kramers doublet ground state, as observed in ODMR.

Such effects in DAP systems have already been reported in wide-band-gap materials such as zinc sulfide[31] and diamond[32], but only phenomenological models have been proposed to explain the observations. Intersystem crossing (ISC) processes for spin pairs have mostly been discussed in the context of organic chromophores[33–37], involving dipolar mixing ($\hat{H}_{di}$), spin-orbit coupling ($\hat{H}_{soc}$), and hyperfine coupling ($\hat{H}_{hf}$).

However, none of these terms can be easily determined in our system due to their dependence on both donor-acceptor distance and the $D_s$–$Q_s$ energy splitting. The $D_s$–$Q_s$ splitting scales with distance as $R^{-6}$. In comparison, ISC mediated by hyperfine interaction decays much more slowly than that caused by dipolar or spin-orbit coupling.

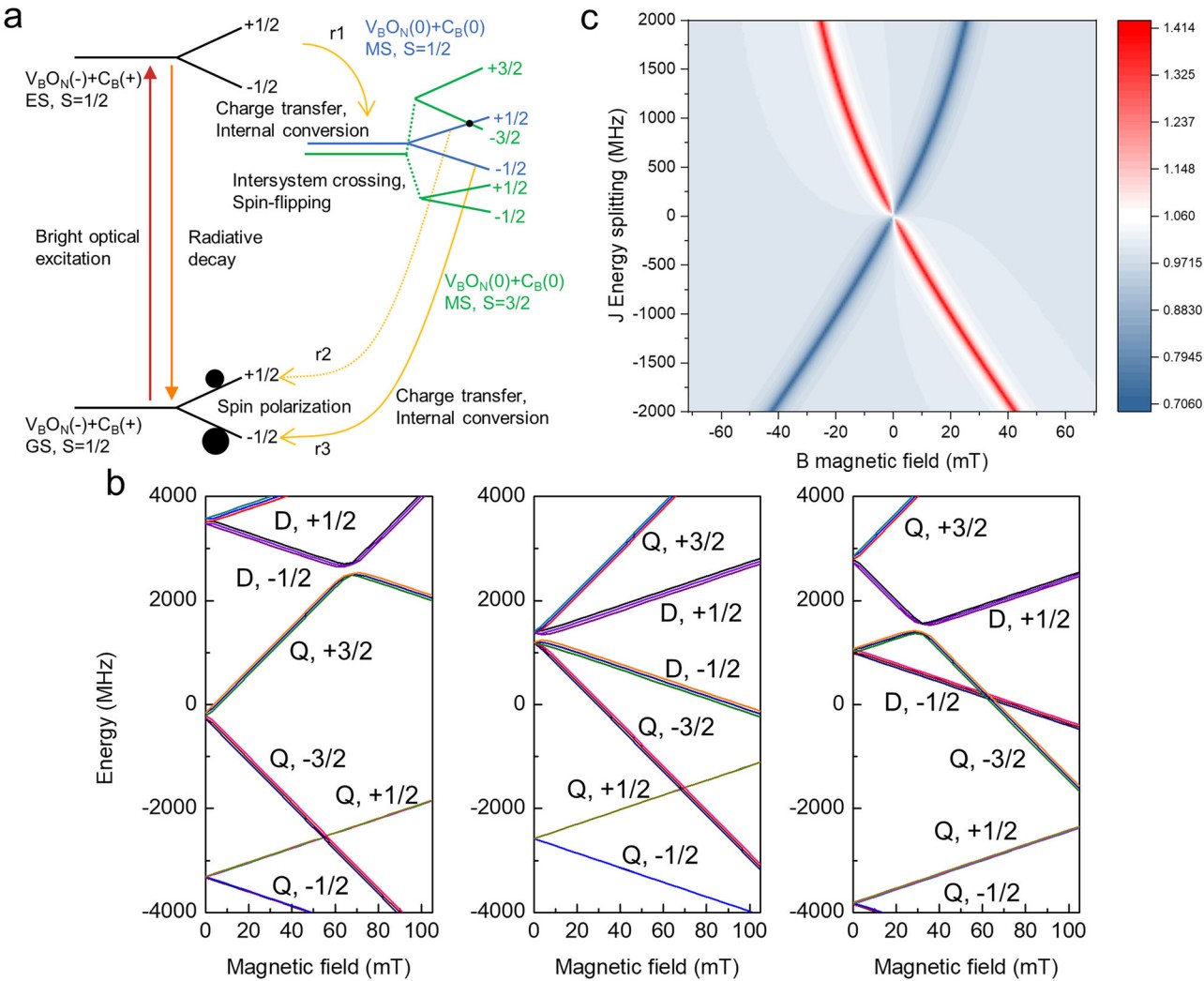

**Fig. 4 | The optical pumping loop of $C_B$–$O_N V_B$. a** Bright excitation (red arrow) from the ground state (GS) to the excited state (ES) is localized on $O_N V_B^-$. Through internal conversion, the ES can relax to the doublet metastable state of $C_B^0$--$O_N V_B^0$, denoted as $D_s$. The $D_s$ level is separated by an energy $J > 0$ from the quartet metastable state $Q_s$ of the same $C_B^0$–$O_N V_B^0$ configuration. The $\pm 1/2$ and $\pm 3/2$ spin sublevels within $Q_s$ are split by zero-field splitting. The black dot indicates spin mixing between $D_s$ and $Q_s$, while the black circle indicates the population magnitude in the ground state. **b** Spin sublevels in $D_s$ and $Q_s$ under magnetic fields, shown for three values of $J$: − 3000, 0, and + 3000 MHz. Source data are provided as a Source Data file. **c** Population ratio between the $|D_s, +1/2\rangle$ and $|D_s, −1/2\rangle$ spin sublevels in the metastable state as a function of applied magnetic field.

For a specific distance $R_i$, the rate of spin-flip can be evaluated using Fermi's golden rule:

$$\Gamma = \frac{2\pi}{\hbar} \left| \langle \psi(D_s) | \hat{H}_{di} + \hat{H}_{soc} + \hat{H}_{hf} | \psi(Q_s) \rangle \right|^2 L(J), \quad (2)$$

where $\psi(D_s)$ and $\psi(Q_s)$ are the doublet and quartet wavefunctions, respectively, and $L(J)$ is the lineshape function associated with the phonon overlap between the two states. Using the strongest hyperfine coupling of 55 MHz arising from the two $^{14}$N nuclear spins of $O_N V_B$ in the $C_B5$ configuration, the estimated rate has an upper bound of $10^8$ Hz, assuming that the doublet and quartet states are nearly degenerate in both energy and geometry. This upper bound may be reduced, for instance, through $L(J)$, but it still represents a sufficiently fast process for efficient spin mixing.

The distance between the DAP constituents should be long enough to reduce the energy gap $J$ between $D_s$ and $Q_s$, but not so long that the spin-mixing rate becomes negligible. Unfortunately, direct calculation of the $D_s$ and $Q_s$ states and their splitting $J$ is beyond the scope of the present study, as it requires multi-reference methods that

are not feasible for the supercell sizes needed to model DAPs. Nevertheless, an effective spin Hamiltonian for the $O_N V_B^0$–$C_B^0$ system, composed of coupled spin pairs with $S_1 = 1$ and $S_2 = 1/2$ in a magnetic field $B$, may be written as

$$H_{spin} = S_1 \mathbf{D} S_1 + \sum_i H_{hf}^{(i)} + g B S_1 + g B S_2 + J S_1 S_2, \quad (3)$$

where the electron gyromagnetic factor is $g \approx 2.0023$ (due to the very weak spin-orbit interaction) for each constituent, $J$ is the isotropic exchange interaction between the spin pairs, $\mathbf{D}$ is the zero-field splitting tensor of $O_N V_B^0$ [$D = 3.8$ GHz and $E = 0.091$ GHz], and we include only the largest hyperfine couplings from the two $^{14}$N ($I = 1$) nuclear spins. We numerically solve Eq. (3) as a function of magnetic field $B$ (applied perpendicular to the hBN sheet) and spin exchange coupling $J$, using the Easyspin software package[38].

Figure 4b shows the splitting of spin sublevels under magnetic field for various $J$ values, which can exhibit different magnitudes and signs depending on the relative orientation of the coupled spins, as observed in our DFT calculations. Due to the zero-field splitting of the

triplet sublevels of $O_NV_B$, the $|Q_s, \pm 1/2\rangle$ levels always lie lower in energy, and the dominant spin mixing occurs between $|Q_s, \pm 3/2\rangle$ and $|D_s, \pm 1/2\rangle$.

The mixing primarily involves $|D_s, -1/2\rangle$ and $|Q_s, +3/2\rangle$ when $J < 0$, while for $J > 0$, it occurs between $|D_s, +1/2\rangle$ and $|Q_s, -3/2\rangle$. Differences in the coefficients of these spin sublevel eigenstates imply variations in spin population, which can lead to ODMR contrast that is tunable via an external magnetic field. More specifically, variation in mixing with the quartet states results in different non-radiative relaxation rates, giving rise to hyperpolarization and detectable ODMR contrast. For each finite $J$-coupling magnitude, we observe a strong dependence of the mixing coefficient ratios on the magnetic field, as illustrated in Fig. 4b. This difference becomes particularly pronounced near the avoided crossing, where the ratio reaches a maximum of approximately 1.4. Ultimately, such asymmetry in mixing is expected to translate into differences in internal conversion rates back to the ground state. Importantly, for $J = 0$−i.e., when the two defects are non-interacting−both doublet states mix equally with the quartet manifold at each magnetic field, resulting in no discernible difference in non-radiative decay.

To further substantiate the role of spin mixing in mediating the ODMR signal, we calculated the internal conversion rates to and from the metastable $D_s$ state. We approach this process by evaluating transitions between the excited state and $D_s$ for $r1$, and between $D_s$ and the doublet ground state for $r2$ and $r3$, respectively. Due to the small transition dipole moment associated with the charge transfer process, as shown in Supplementary Fig. 4, the non-radiative decay channels are expected to dominate. At short donor-acceptor distances below 1 nm, the small energy difference between the optically active doublet excited state and the dim metastable state, combined with strong electron-phonon coupling, leads to a fast internal conversion rate on the order of $10^{10}$ MHz. Consequently, the system relaxes to the ground state without producing a detectable optical signal through the dim state. The orbital overlap integral−and the resulting electron-phonon coupling−decays with increasing distance. The $R$-dependent value of the overlap integrals can be approximated by the overlap of two Slater orbitals, which scales as $e^{-R}$ [39]. Using two configurations, $C_B5$ and $C_B7$, we extrapolate the electron-phonon coupling and internal conversion rate to longer donor-acceptor separations that cannot be directly computed using DFT, as discussed in Supplementary Note 3. The estimated internal conversion rate $r1$ is approximately 6-8 MHz at an 18 Å separation. This yields competitive radiative and non-radiative pathways from the optically active excited state of the $C_B-O_NV_B$ DAP defects. Accordingly, the ZPL wavelength, quantum yield, and brightness of these DAP defects are expected to depend on the actual donor-acceptor distance. Notably, both the weak radiative and the non-radiative ($r3$) internal conversion pathways connect the metastable state to the doublet ground state (see Supplementary Note 3), thereby completing the optical spin-polarization loop in these DAP systems.

With around 2 nm separation between $C_B$ and $O_NV_B$, the present coupled spin pair model may account for the previously reported ODMR signals originating from ground-state $S = 1/2$ defects in hBN sheets and nanotubes[12,40,41]. We underline that if the metastable state is sufficiently long-lived to carry out spin operations (e.g., ST1 defect in diamond[42]), then this spin-pair model can be extended to ODMR centers in hBN where spin polarization is observed in the metastable state. In this case, the ground state may be a singlet, formed by an $S = 1/2$ acceptor and an $S = 1/2$ donor, whereas the metastable states comprise singlet and triplet spin pairs. We note recent works[40,43,44] published during the preparation of our manuscript, which observe this phenomenon without providing microscopic models for the underlying processes. Indeed, other types of coupled spin pairs may exist in hBN beyond those discussed in detail in our study. In Supplementary Note 4, we examine additional DAP configurations, such as $C_B-V_B$ and $C_B-C_NV_B$.

The $C_B-V_B$ pair represents a prototype of an $S = 1$ ground-state system with triplet and singlet spin pairs in the metastable state, while $C_B-C_NV_B$ is an example of a system with an $S = 1/2$ ground state. The latter is similar to the $C_B-O_NV_B$ configuration but exhibits distinct magneto-optical characteristics. We also note that the previously studied $C_B-C_N$ DAPs[30] serve as prototypes of an $S = 0$ ground state, with triplet and singlet spin pairs emerging in the metastable state. These findings demonstrate that our basic DAP model offers a new pathway for defect spin identification in hBN.

### Discussion on color center identification

Generally, the atomic structure of optical emitters in hBN is challenging to determine experimentally. Direct imaging through high-resolution transmission electron microscopy can provide geometric information about defects on the top layer of hBN[45,46]. However, the optical signals sometimes originate from defects located deeper within the crystal.

EPR is a powerful tool for identifying paramagnetic point defects (with one or more unpaired electrons), based on their zero-field splitting (ZFS) and hyperfine interactions. In hBN, the interpretation of EPR signals is challenging but feasible in principle[23,47], because both boron and nitrogen have 100% natural abundance of non-zero nuclear spin isotopes−a factor that increases the complexity of extracting the appropriate spin Hamiltonian. Furthermore, the ZFS of high-spin defects provides an important parameter for theoretical benchmarking[19,48]. However, EPR cannot detect non-magnetic defects, and the observed signals may originate from multiple overlapping defect types.

Confocal microscopy and photon antibunching measurements can offer optical insights, such as ZPL energy, PSB, and fluorescence lifetime of individual defects. Still, distinct defects may produce similar optical features, leading to potential misassignments. By applying external fields−such as strain[49] or electric fields[13,50]−the optical response of the emitters can be modulated, offering additional information for defect identification. In our model, we propose that the observed optical signals could originate from spin pairs rather than isolated single defects. Multiple optical transition pathways within such spin-pair systems can lead to spin polarization and ODMR contrast, providing another route for identifying quantum defects.

In summary, a comprehensive analysis of SPEs or qubits in hBN requires a multi-dimensional approach that integrates structural, magnetic, optical, and field-dependent characterizations. Our study shows that some observations can be interpreted as direct interaction between defect pairs which underscores the complexity of defect identification of emitters and qubits in wide-band-gap materials such as the 2D hBN.

## Conclusion

In summary, we propose that DAP emission is responsible for the visible ZPL emission at around 2.0 eV. The variance in ZPL wavelengths and optical lifetimes reported in experiments can be explained by the type and spatial separation of the donor defect relative to a key acceptor defect, $O_NV_B$. Our simulations reveal the ODMR mechanism associated with spin pairs in certain DAP structures, which are activated via dim metastable states and manifest as ODMR signals from an $S = 1/2$ ground state under non-zero constant magnetic fields. Our model also explains the challenge of producing indistinguishable SPEs in hBN, as the spatial arrangement of defect pairs must be precisely engineered. DAPs are common in wide band gap semiconductors and can account for both optical emissions and ODMR signals in Kramers doublet spin systems. The general mechanisms proposed here offer insight not only into defect engineering in hBN for quantum information processing but also into broader optoelectronic applications involving defects in other semiconductors.

## Methods

### Density functional theory calculations

In this paper, we performed density functional theory (DFT) calculations using the Vienna Ab Initio Simulation Package (VASP) code[51,52] with a plane-wave basis set. We applied a plane-wave cutoff energy of 450 eV, and the convergence test is provided in Supplementary Note 6. The valence electrons and the core regions were described using the projector augmented wave (PAW) potentials[53,54].

A $8 \times 8$ two-layered supercell was used to avoid interactions between periodic images, and this size was sufficient to apply the $\Gamma$-point sampling scheme. The interlayer van der Waals interactions were described using the DFT-D3 method of Grimme[55].

Using a mixing parameter of $\alpha = 0.32$, the hybrid functional of Heyd, Scuseria, and Ernzerhof (HSE)[56] reproduces the experimental optical band gap of approximately 6 eV (excluding electron-phonon renormalization effects). This functional was used for geometry optimization and electronic structure calculations. The convergence threshold for atomic forces was set to 0.01 eV/Å.

Electronic excited states were calculated using the $\Delta$SCF method[57]. Band alignment in charge correction was performed based on the core level energy of a selected atom located far from the defect center, both in the defective and perfect supercells.

The zero-field splitting due to dipolar electron-spin interaction was calculated within the PAW formalism[58], as implemented in VASP by Martijn Marsman. We applied spin decontamination by averaging values over two symmetric spin-0 configurations to arrive at the final results[59]. The hyperfine tensor was calculated with the Fermi-contact term included, considering only the major contribution from the two nitrogen atoms[60].

For the electron-phonon coupling, we first generated configuration coordinate diagrams of the relevant states, then computed the electron-phonon matrix elements using wavefunction overlap and linear response theory. Non-radiative decay rates were calculated using Fermi's golden rule, as implemented in the NONRAD code[61].

## Data availability

The data that support the findings of this study are available from the corresponding author upon request. Source data are provided with this paper.

## Code availability

The codes that were used in this study are available upon request to the corresponding author.

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

## Acknowledgements
Support by the Ministry of Culture and Innovation and the National Research, Development and Innovation Office within the Quantum Information National Laboratory of Hungary (Grant No. 2022-2.1.1-NL-2022-00004) as well as the European Commission for the projects QuMicro (Grant No. 101046911) and SPINUS (Grant No. 101135699) are much appreciated. A.G. acknowledges the high-performance computational resources provided by KIFÜ (Governmental Agency for IT Development of Hungary). A.P. acknowledges the financial support of Janos Bolyai Research Fellowship of the Hungarian Academy of Sciences.

## Author contributions
S.L. and A.G. designed the project. S.L. and A.P. carried out the DFT calculation. All authors discussed the results and contributed to the manuscript writing. A.G. led the scientific project.

## Funding

## Competing interests
The authors declare no competing interests.
