## [Transparent Peer Review file · Nature Communications]

Quantum Emission from Coupled Spin Pairs in Hexagonal Boron Nitride

Corresponding Author: Professor Adam Gali

Version 0:

Reviewer comments:

Reviewer #1

(Remarks to the Author)

Despite years of intensive research, the origin of emissions in hBN characterized by a ZPL of ~ 2.0 eV is still unclear. This manuscript proposes a new model that not only explains the optical features of these emissions but also accounts for the properties of the experimentally observed ODMR signal. I find the results presented in this paper to be sound, robust and deserving of publication in Nature Communications. The methodology of the study is convincing and solid. The paper is well-written and structured appropriately.

Before publication, I would like to request that the following questions be addressed:

1. The proposed mechanisms of interaction with photons or the magnetic field would only be possible under one condition: that CB is positively charged, and VBON is negatively charged. However, based on the calculations by Weston et al. (Phys. Rev. B 97, 214104 (2018)), this condition holds for a relatively narrow range of the Fermi level, specifically between 2.44 and 3.71 eV above the VBM. This range corresponds to intrinsic hBN samples. The presence of impurities or specific growth conditions could easily shift the Fermi level outside this range. Could you please comment on this? I would expect that the stability range of the CB⁺-VBON⁻ pairs could be broader than 2.44-3.71 eV, since their Coulombic attraction would reduce their formation energy, shifting the charge-state transition levels towards the band edges. However, this would require further clarification.
2. In the manuscript, it is stated: 'In summary, we propose that the DAP emission is responsible for most of the visible ZPL emission at around 2.0 eV.' However, many experiments have shown a correlation between the number of visible emitters and the presence of carbon in the sample (e.g., Mendelson et al., Nat. Mater. 20, 321 (2021)). The ON⁺-VBON⁻ pair does not contain carbon, whereas the CB⁺-VBON⁻ pair does. However, carbon is not required for emission. If I understand correctly, the presence of carbon affects the electronic structure of the CB-VBON pairs but is not essential for the emission itself, as pure VBON⁻ would also be an emitter with ZPL around 2.0 eV. How do you explain the experimentally observed correlation between the presence of visible emitters and the carbon content? If the DAP pairs you propose account for only a subset of the emitters, the claim that they are responsible for most of the emissions may be an overstatement.
3. Regarding Supplementary Figure 4 – which value of epsilon was used in this calculation?
4. I request a clear introduction and explanation of the meanings of the a1, a2, b1, and b2 orbitals in the main text.

Reviewer #2

(Remarks to the Author)

I attach the review in a separate file

Reviewer #3

(Remarks to the Author)

I have written my reviewer's report in a pdf file. Please see attachment.

Reviewer #4

(Remarks to the Author)

Version 1:

Reviewer comments:

Reviewer #1

(Remarks to the Author)

I maintain the opinion I expressed in my first review that the presented results are robust, interesting, and important. Moreover, after revision, they are now presented in a much clearer and more accessible manner. Honestly, I found the revised manuscript much more enjoyable to read compared to the initial version.

I recommend the manuscript for publication.

The only remark I would like to make is that the formation energy of CB-VBON and ON-VBON depends on the chemical potential during growth (e.g., N-poor or N-rich conditions). Therefore, it should be clarified to which specific conditions the presented values correspond.

Reviewer #2

(Remarks to the Author)

I thank the author for the response.

I do not have additional questions, the authors addressed all my concerns and replied to my questions, so I recommend the paper for publication.

Reviewer #3

(Remarks to the Author)

The authors have fairly answered my questions. I recommend the manuscript to be published in Nature Communication after these minor revisions.

1. Eq.(2) should be checked on the subscripts. I would also request that each Hamiltonian be stated explicitly.
2. The dielectric constant value should be stated in the manuscript.
3. There are several awkward sentences or phrases in the Supplementary (e.g. first line below Section V, also a few lines after that).
4. Please provide citation that IC is significantly faster than ISC.
5. I would request that the abstract be revised and added more information about the results to reflect the revised manuscript. In the current version, only one sentence in the abstract is about the results.

Reviewer #4

(Remarks to the Author)

ANSWER TO REFEREES

We thank all the Referees for their reports. We provide below a detailed answer. The major changes are indicated in the highlighted PDF file provided in the resubmission. We further polished the text accordingly to increase the clarity and readability.

Reviewer #1 (Remarks to the Author):

Despite years of intensive research, the origin of emissions in hBN characterized by a ZPL of ~ 2.0 eV is still unclear. This manuscript proposes a new model that not only explains the optical features of these emissions but also accounts for the properties of the experimentally observed ODMR signal. I find the results presented in this paper to be sound, robust and deserving of publication in Nature Communications. The methodology of the study is convincing and solid. The paper is well-written and structured appropriately.

Before publication, I would like to request that the following questions be addressed:

1. The proposed mechanisms of interaction with photons or the magnetic field would only be possible under one condition: that C_B is positively charged, and V_{BON} is negatively charged. However, based on the calculations by Weston et al. (Phys. Rev. B 97, 214104 (2018)), this condition holds for a relatively narrow range of the Fermi level, specifically between 2.44 and 3.71 eV above the VBM. This range corresponds to intrinsic hBN samples. The presence of impurities or specific growth conditions could easily shift the Fermi level outside this range. Could you please comment on this? I would expect that the stability range of the $C_B^+ - V_{BON}^-$ pairs could be broader than 2.44-3.71 eV, since their Coulombic attraction would reduce their formation energy, shifting the charge-state transition levels towards the band edges. However, this would require further clarification.

We thank the Reviewer for the insightful comment. Indeed, as the Reviewer mentioned, there is a Coulombic attraction between C_B and V_{BON} defects, so that the total energy decreases when the two defects are close to each other, as shown in Figure 1 below. In response to this question, we have also calculated the formation energies for the $C_B + O_N V_B$ pairs, shown in Figure 2. As the Reviewer expected, the charge transition levels (CTL) differ across various configurations. However, the neutral charge state extends over a broader region than the 2.44-3.71 eV span, computed for the isolated defects. This leads us to conclude that the Coulombic attraction enhances the stability of the $C_B^+ + O_N V_B^-$ pairs over a wider range of Fermi energies. These results are now added as Supplementary Figure 3 and discussed in main text.

Figure 1. The HSE total energy difference of various $C_B + O_N V_B$ pairs.

Figure 2. The formation energy of $C_B + O_N V_B$ pairs.

CTL	(-1/0)	(0/+1)
C_{B3}	1.54	3.83
C_{B8}	2.38	4.20
C_{B4}	1.71	3.66

Table 1. The CTL of $C_B + O_N V_B$ pairs.

2. In the manuscript, it is stated: 'In summary, we propose that the DAP emission is responsible for most of the visible ZPL emission at around 2.0 eV.' However, many experiments have shown a correlation between the number of visible emitters and the presence of carbon in the sample (e.g., Mendelson et al., Nat. Mater. 20, 321 (2021)). The $ON^+ - VBON^-$ pair does not contain carbon, whereas the $CB^+ - VBON^-$ pair does. However, carbon is not required for emission. If I understand correctly, the presence of carbon affects the electronic structure of the $CB - VBON$ pairs but is not essential for the emission itself, as pure $VBON^-$ would also be an emitter with ZPL around 2.0 eV. How do you explain the experimentally observed correlation between the presence of visible emitters and the carbon content? If the DAP pairs you propose account for only a subset of the emitters, the claim that they are responsible for most of the emissions may be an overstatement.

We apologise for the misleading statement in the conclusion part. We agree that it is not appropriate to say that the DAP emission is responsible for most of the visible ZPL emission at around 2 eV and

we updated the sentence in conclusion part.

In fact, we propose that the DAP emission as a promising candidate for the visible ZPL emission at around 2.0 eV, especially for the $S = 1/2$ defect with an ODMR signal. Indeed, as the Reviewer mentioned, there are studies indicating the visible single photon emission originates from isolated defects. In contrast, our main claim is that the DAPs provide a general mechanism, which represents a possible explanation for the observed ODMR signal from the spin-1/2 systems.

3. Regarding Supplementary Figure 4 – which value of epsilon was used in this calculation?

The static in-plane dielectric constant 6.93 is used to do the calculation.

4. I request a clear introduction and explanation of the meanings of the a_1 , a_2 , b_1 , and b_2 orbitals in the main text.

The negatively charged $O_N V_B^-$ defect has C_{2v} symmetry, so we labelled the irreducible representation of localized wavefunctions of defect as shown in a following figure. The symmetry operations include C_2 , $\sigma_v(xz)$ and $\sigma_v(yz)$.

Figure 3. The electronic structure and localized states of $O_N V_B^-$ at ground state.

C_{2v}	E	C_2	$\sigma_v(xz)$	$\sigma_v(yz)$
A_1	1	1	1	1
A_2	1	1	-1	-1
B_1	1	-1	1	-1
B_2	1	-1	-1	1

Table 2. The character table of C_{2v} symmetry.

According to the character table of C_{2v} symmetry, the defect levels can be labelled as a_1 , a_2 , b_1 , and b_2 based on the symmetry operations of the wavefunction. We updated the description in the main text as requested by the Reviewer.

Reviewer #2 (Remarks to the Author):

The paper “Quantum Emission from Coupled Spin Pairs in Hexagonal Boron Nitride” by Li et al investigate the donor-acceptor pair composed by the primary acceptor defect $O_N V_B^-$ and surrounding donor ON, CB. The authors discuss the energy diagram related to various transitions and charge states as a function of donor-acceptor separation. The ODMR mechanism is qualitatively explored in relation to energy diagrams and relative recombination rates. The authors suggest that

the spin mixing of two metastable states could explain the observed spin polarization and ODMR contrast, with transition rates being sensitive to donor-acceptor separation and potential engineering. Below are a few points I found unclear and believe could benefit from clarification:

1. The work considers the charged donor-acceptor pair and the neutral donor acceptor pair as two distinct states. For instance, in the statement, “Furthermore, the total energy of $CB^+ - ONVB^-$ configuration is lower than that of $CB^0 - ONVB^0$ ”, the energy difference between the two configurations is compared. It remains unclear how the total energy is calculated for systems with and without local charge transfer. Is it determined by DFT total energy computations for different electronic configurations based on the atomic model shown in Figure 1a? Additionally, is this claim valid for all 10 configurations proposed in the main text?

For this statement, the total energy of the pair is simply the summation of two isolated defects. So here $E(C_B^+) + E(O_N V_B^-)$ is lower than $E(C_B^0) + E(O_N V_B^0)$. The calculation does not contain the interaction between the two defects, or the local charge transfer.

With the DAP model that two defects are coexist in the supercell model, we could calculate the total energy with the charge transfer included. The charge transfer process is considered in the manuscript that we calculate the metastable state as shown in Figure 4a. At least with short distance considered in our manuscript, where the charge transfer could happen, one electron from C_B always transfers to $O_N V_B$ and this is the ground state and valid for all 10 configurations proposed in the main text. Supplementary Figure 4 also shows the energy of metastable state $E(C_B^0 + O_N V_B^0)$ aligned to ground state $E(C_B^+ + O_N V_B^-)$. However, as we estimated in discussion part, with the increase of distance, the charge transfer from C_B to $O_N V_B$ might not occur due to the decrease of wavefunctions overlap.

2. The bright ZPL emissions are attributed to the $a1$ to $b1$ transition from the $(ONVB)^-1$ site. The paper also mentions that the charge transfer process from $(ONVB)^-1$ to CB^+ can also luminesce, resulting in two metastable states of $S=1/2$ and $S=3/2$. Supplementary Figure 3 depicts the luminescence energy for the ground state to the $S=3/2$ metastable state and compares it to the ZPL energy of the internal $a1$ to $b1$ transition (main text Figure 1b). However, there are two points of confusion:

a. The supplementary figure denotes energy as the “energy difference of Kohn-Sham levels,” which typically refers to the single-particle energy level differences. Meanwhile, the main text states, “Supplementary Figure 3 shows the Kohn-Sham energy difference,” suggesting the total energy difference between the ground state and excited state configurations. If it indeed refers to the total energy difference, why is there such a significant difference between the bright transition energy in Supplementary Figure 3 and the ZPL energy in Figure 1b?

Indeed, as the Reviewer mentioned, the energy difference of Kohn-Sham levels refers to the single-particle energy level differences. Simple ground state calculation produces this result. However, this is not the total energy difference between the ground state and excited state associated with the ZPL energy as observed in experiments.

At most cases, the single-particle energy level difference is much larger than the calculated ZPL since the optical excitation includes the relaxation of the atoms and rearrangement of the electron density, and the excitonic effect. These effects lower the total energy of the excited state compared to the simple KS eigenvalue difference. We updated the energy difference for these states in

Supplementary Figure 4.

b. The distinction between the ZPL associated with charge-transfer-related transitions and the internal $a_1 \rightarrow b_1$ transition at $(ONVB)-1$ is unclear. If both are derived from constrained DFT with different single-particle level configurations of the same DAP model, it would be helpful for the authors to clarify the transitions or configurations in the single-particle diagram. Including an electronic configuration diagram or group theory analysis would enhance understanding and distinguishment of the metastable $S=1/2$ and $S=3/2$ states, as well as the ground, excited, the dark transition, dim transition in Supplementary Fig3.

We thank the Reviewer for the constructive comment. First, we changed the “dark transition” to “dim transition” in Supplementary Figure 4 to be consistent with the main text.

In the following figure, we show the single-particle level ground state, excited state, and the two metastable states with the C_B4 configuration. To clearly identify the occupation of the orbitals, we use the single-particle level at ground state as a reference, since the excitation changes the order of orbitals and makes them difficult to distinguish for the readers.

So, at the bright excited state (ES), the electron occupies the b_1 orbital from $O_N V_B$; at the dim metastable state (MS, 1/2), the electron transfers back to C_B from $O_N V_B$ in a spin-conserving way; at the other metastable state (MS, 3/2), the electron transfers back to C_B from $O_N V_B$ and flips the spin direction.

Figure 4. The electron occupation of C_B4 at ground state (GS), excited state (ES), metastable state (MS). Filled and unfilled triangles indicate the empty and occupied defect levels in gap and the direction of triangle denotes the spin majority or minority. The red triangle is the a_1 orbital of $O_N V_B$ that considered to excite. The solid and dash orange lines represent the bright and dim excitation paths.

3. Equation (1) Model and Energy Interpretation: Equation (1) aims to explain the emission mechanism related to charge-transfer luminescence. However, it lacks sufficient explanation of its physical meaning, particularly regarding the definition of $E(Ri)$ represents the emission energy, the substantial energy differences between Supplementary Figures 3 and 4 (dark transitions) require further discussion and explanation. For example, in the statement, “The total energy of $S = 3/2$ state lies with 1.68 eV to 2.36 eV higher than that of the ground state depending on the DAP distance and this means that the bright excited state (~ 2.0 eV) and the metastable states can switch their energy position.” the origin of these energy values is unclear.

$E(R_i)$ is the transition energy of the donor-acceptor model, that the electron transfers from $O_N V_B^-$ back to C_B^+ and the DAP becomes $(O_N V_B^0) + (C_B^0)$. E_{gap} is the bandgap energy, and E_D and E_A are the charge transition levels calculated based on previous studies [Phys. Rev. B 97, 214104 (2018)]. R_i here is the distance between donor and acceptor, and defined as the distance between oxygen atom in $O_N V_B$ and the donor atom (carbon or oxygen).

As we can see from the equation, the energies are based on the total energy of the optimized excited state and ground state, therefore the calculated energy $E(R_i)$ is closer to the ZPL simulated in the main text, quantitatively, as shown in Supplementary Figure 4. Similar to above comment from the Reviewer, the substantial energy difference between Supplementary Figures 4 a,b and c,d is because the energy difference is referred to the single-particle Kohn-Sham orbitals. We slightly modified Supplementary Figure 4 to distinguish the energy there from other figures. The y axis is labelled as “Kohn-Sham energy difference” and the red line indicates the dim transition MS, 1/2 state.

Figure 5. The Kohn-Sham energy difference of the bright and dim transition ($S=1/2$) of $C_B+O_N V_B$.

We further show the (a) Kohn-Sham energy difference and (b) transition energy of dim MS, 3/2 state in Figure 6. $S = 3/2$ could converge properly in our calculations while $S = 1/2$ cannot so we use $S = 3/2$ to estimate the energy of $S = 1/2$. This is a reasonable approximation, since the energies of the two states differ by the exchange contribution ($3J$), which rapidly decays with the separation distance. From (b) we can get the “The total energy of $S = 3/2$ state lies with 1.68 eV to 2.36 eV higher than that of the ground state”. The figure was added to Supplementary Figure 4. We also updated the discussion in the main text.

Figure 6. The Kohn-Sham energy difference and the optimized transition energy of the dim transition ($S = 3/2$).

4. Consistency Between Figures and Models: In the discussion, Figures 3b and 3c present a donor-acceptor model. It would be beneficial to explicitly connect these states to the energy diagram in Figure 4a for improved clarity.

Thank you for the comment, we have updated the discussion of Figure 3 and 4 accordingly. Figures 3b and c provide the charge transfer between donor and acceptor and show the corresponding fluorescence. In our model, the electron moves from C_B to $O_N V_B$ and make $O_N V_B$ negatively charged (A^-) with positively charged C_B (D^+). The charge transfer from $O_N V_B^-$ to C_B^+ can be described by the diagram in Figure 3b (this is the dim transition discussed in main text). However, the two cases in Figure 3b and c overlook the possibility of so called “local excitation” or bright transition, where the electron is not excited back to the donor (D^+) but is trapped by the acceptor (A^-). In other words, the optical transition solely occurs within a single defect, since the dim and bright transitions have close energy, but the latter one has a larger wavefunction overlap.

In Figure 4, the discussion is based on the DAP pairs so the states can be described as following figure that the ground and excited states are $A^- + D^+$ and the MSs are $A^0 + D^0$.

Figure 7. The states described by the DAP model in Figure 3 in the main text.

5. Radiative and Non-Radiative Processes: The transition $r2$ and $r3$ account for both radiative and non-radiative processes. However, their discussion appears loosely organized. For instance, the radiative transition dipole moment is computed in the Results section, while non-radiative lifetimes (internal conversion rates) are only discussed towards the end of the Discussion section. Since energy level differences influence transition rates, corresponding data could be more clearly presented. Specific details on how internal conversion rates were calculated should be provided, along with citations. A table summarizing the energy levels of the ground state (GS), excited state (ES), and metastable states (Ds and Qs) would be very helpful. Additionally, the method for extracting electron-phonon coupling requires further explanation.

Thank you for the constructive comment. The transition $r2$ and $r3$ account for both radiative and non-radiative processes for different spin-sublevels. What we calculated and presented in Figure 3 is the radiative transition dipole moment of the local bright transition. It corresponds to the orange solid straight line in Figure 4a. We modified the related description in the main text to clarify this point. We put it in Figure 3 because the fast convergence of radiative transition dipole moment is

consistent with the ZPL change, shown in Figures 1 and 2. We rewrite the discussion towards the radiative and nonradiative process on Page 13-14.

The mentioned “weak radiative decay pathway” in the discussion section refers to the radiative transition of the dim transition, which is weak due to a charge transfer character. Initially, we did not provide the transition dipole moment for the dim transition at the main text. Now, we plot it below and add it to Supplementary Figure 4. As mentioned in the main text, the donors (C_B and O_N) have empty out-of-plane p_z orbitals in the gap so the overlap of wavefunction between the occupied a_1 in $O_N V_B$ and the p_z orbital is small. The transition dipole moment is much smaller than those for the bright excitation in Figure 3.

Figure 8. The transition dipole moment of dim transition ($S = 1/2$).

We have also plotted the total energy difference for the excited state (ES) and metastable state (MS, $S = 3/2$) with respect to the ground state (GS) energy. We only plot the $S = 3/2$ state energy, since the $S = 1/2$ state should have very a close total energy. All these figures are now included in Supplementary Figure 4. This energy difference between the metastable states is estimated in Figure 4c with J , which is a much smaller value compared to the other energy differences.

Figure 9. The optimized transition energy of the bright and dim transition ($S = 3/2$).

To calculate the electron-phonon coupling, the first step was to get a configuration coordinate diagram (CCD) with the relaxed ground and excited state geometries. Based on the potential energy surface, it is possible to extract the phonon frequencies and the ΔQ which is the difference in the equilibrium geometries of the initial and final state along the configuration coordinate. The electron phonon matrix element can be evaluated within the PAW formalism based on the linear response

theory. It could be written as $W_{if} = (E_f - E_i)\langle\varphi_i(0)|\delta\varphi_f(Q)\rangle$, $\langle\varphi_i(0)|\delta\varphi_f(Q)\rangle$ is a wavefunction overlap function. W_{if} is evaluated by computing the slope of $\langle\varphi_i(0)|\delta\varphi_f(Q)\rangle$ as a function of the Q and is associated with the defect levels involved in the excitation. In the manuscript, we have considered the a_1 , b_1 orbitals from O_{NV_B} and the state from C_B . The calculated result is shown in Table II in Supplementary Note 3. With these parameters, it is possible to calculate the nonradiative decay process with Fermi golden rule with the following equation:

$$\Gamma_{non} = f \frac{2\pi}{\hbar} W_{if}^2 \sum_m \omega_m \sum_n |\langle\chi_{im}|Q - Q_0|\chi_{fn}\rangle|^2 \times \delta(\Delta E + m\hbar\Omega_i - n\hbar\Omega_f).$$

Ω_i and Ω_f are the harmonic phonon frequencies of the initial and final state, and ΔE is the energy difference between them. ω_m is the thermal occupation factor for the m th vibrational level in the initial state. The details can be found in [Comput. Phys. Commun. 267, 108056 (2021)].

Figure 10. The CCD and electron phonon matrix element of the defect levels that are responsible for $r3$ process with C_{B7} .

As we mentioned in Supplementary Note 3, the electron-phonon coupling is distance-dependent so we extrapolate the coupling strength exponentially based on the above result in a finite supercell. With the fitted equation, the electron-phonon coupling is estimated in Supplementary Figure 6. We included brief discussion in the method part.

6. The following conclusion appears inconsistent: “the small energy difference between the optically active doublet excited state and the dim metastable state and the strong electron-phonon coupling lead to fast internal conversion rate around 10^{10} MHz. Therefore the system relaxes back to ground state rapidly without detectable optical signal.” While the first part asserts that the transition from the excited state (ES) to the dim state (Ds) is rapid (r1), the subsequent statement focuses on the fast and fully non-radiative relaxation from Ds to the ground state (r2,r3). The reasoning appears inconsistent and could be clarified. For these reason we cannot recommend publication right now and the manuscript requires revision.

Indeed, as the Reviewer said the first part provides the conversion rate from ES to dim state with $r1$. The latter sentence is based on the weak radiative transition as discussion above. Beside the nonradiative decay from dim state to GS, there is also a radiative decay pathway from the ES, which is slower than the nonradiative process. We write “without detectable optical signal” due to the small transition dipole moment calculated above.

We further calculate the radiative decay from ES to the dim state and from dim state to GS with C_{B5} as an example. The radiative transition can be estimated by:

$$\Gamma_{rad} = \frac{n_D E_{diff}^3 \mu^2}{3\pi\epsilon_0 c^3 \hbar^4}.$$

n_D is the refractive index of hBN at $E \approx 2$ eV. Since the transition dipole moment of the dim state is only 0.16 Debye, the estimated radiative decay rate is in kHz region or less with the input transition energies from Table II. Therefore, the radiative decay would be much slower than the nonradiative process.

What we want to address here is that the small energy gap and the electron-phonon coupling could drive the population transfer from the optically allowed excited state to the dim metastable state. Furthermore, in the dim metastable state, the possible mixing between the spin sublevels ($M_S = 3/2$ and $1/2$) could lead to an inequivalent population of $M_S = \pm 1/2$. By including the radiative relaxation from M_S to the ground state, the inequivalent population still exists and is enhanced by external magnetic field. Through this mechanism, it is possible to obtain an ODMR contrast signal with the spin-1/2 system.

Reviewer #3 (Remarks to the Author):

The authors related and explained the variance in the optical spectral, optical lifetimes and spectral stability of quantum emitters (QE) to donor-acceptor pairs (DAP) in hexagonal boron nitride (hBN). They found that DAPs can exhibit optically detected magnetic resonance (ODMR) signals for the acceptor counterpart of the defect pair with $S = 1/2$ ground state at non-zero magnetic fields depending on the donor partner, and claimed that the DAP model and its transition mechanisms provide a recipe towards defect qubit identification and performance optimization in hBN. General Comments Identifying a microscopic origin of single photons or optically addressable qubits, especially ones that operator at room temperature, is immensely important and has far-reaching impact in quantum information processing. Many techniques have been employed, from which we can obtain different physical quantities. However, it is not clear which property is a smoking gun to identify a qubit origin. This research topic is very active and of high interest to the quantum technology community. The article is well-organized, well-written, comprehensible and easy to follow and contributes significantly to a development in the field. More importantly, it fills in the literature gap for explaining the 2-eV emission from first-principles, rather than phenomenologically. Given the relevance and timely research of the topic, I would recommend the article be published for the community. However, in my opinion, the article is still lacking the novelty, and it should include more DAP systems to validate the claimed results. On the novelty argument, DAP has been reported before, and well documented, as pointed out in many references by the authors (e.g. Phys. Rev. B (2021), 104, 075410) and also Nano Lett. (2022), 22, 1331-1337. Therefore, I would recommend that the article be revised, taking the following comments into account. More essentially, the authors should explain why other mechanisms of photon emission at around 2 eV must be ruled out, and include more results from other DAP systems.

Comments on Content

1. The authors claimed that the variance in the optical properties comes from the type and location of the donor partner. How can they rule out effects of strain, or other mechanisms?

We respectfully disagree with the Reviewer's claim about novelty as we are not aware of any microscopic model that have been proposed to explain the ODMR of Kramers-doublet spins in hBN and its potential connection to DAP. The novelty does not lie in the connection of single photon emitters and DAP but rather (a) the microscopic mechanism of optical spinpolarization and readout mechanism and (b) first-principal calculations and simulations to quantify the respective rates for

certain DAP models that verified our theory.

Nevertheless, we agree with the Reviewer that strain and other mechanisms like local dielectric environment difference can shift the position of ZPL. There are several studies, including also our previous investigation, indicate that strain, polytypes, electric field or stacking fault can cause variance of optical properties like ZPL, PSB and etc.

Here we do not intend to rule out these effects. Instead, we propose yet another possible reason which is not systematically considered. Previous studies usually focus on single defect model to explain the observed phenomenon. By contrast, we wish to address the importance of the defect-defect interaction for the optical properties of single-photon emitter and qubits.

2. It is well known that optical signals alone can provide insufficiently information on the chemical nature of a defect, as local environment (e.g. strain, isotopes, charges) can affect the PL spectra. The authors should clearly explain why their method can uniquely determine a microscopic origin of a qubit in hBN.

Indeed, as the Reviewer mentioned the optical spectra do not typically suffice to determine the chemical nature of a defect. Also, as mentioned above, the local environment could remarkably modify the PL spectra. Our previous study based on a hyperfine analysis (or isotopes effect) together with recent experimental observations provide evidence that the oxygen-related defect exists and gives rise to a visible single photon emission at around 2 eV. We do not claim that we uniquely identified the microscopic origin of the recently observed single-spin ODMR centers as insufficient number of data is available from experiments that should be reproduced to be certain that they observe the same ODMR centers. What we want to address is the mechanism behind the optical spin-polarization and readout of Kramer's doublet-spin defects in hBN which have been experimental observed but not explained.

3. Since the authors claimed that DAP emission is responsible for most of the visible ZPL emission at around 2.0 eV, and DAPs are quite common structures, they should validate their results with more systems other than the ON-ONVB and CB-ONVB systems. Could the authors include results from more DAP systems to validate their model? Maybe in the supplementary material.

We thank the Reviewer for a useful comment. First, we have clarified the claim about "most of the visible ZPL emission." Second, we have considered a DAP of $C_B + V_B$ system, given that the V_B is a common acceptor defect in hBN. To this end, we have simplified the calculated ground states of various configurations by the PBE functional, as shown in Figure 11. Note that the energy levels are not aligned with the VBM. For all these configurations, an electron from C_B is transferred to the V_B , which makes the V_B negatively charged. It is easy to identify the state from C_B^+ (all of them are empty), highlighted by the red color, as well as the empty e-states from V_B^- (blue color). When the two defects are close to each other, the geometry distortion splits the degenerate e-states of V_B^- . As the distance increases, the C_B levels shift down in energy, while e-states shift upward. With the HSE functional, we see the crossing of energy levels from the two defects. Similar to $C_B + O_N V_B$, there are two possible spin-conserving excitations here. The difference is that the local excitation in V_B is also a dim transition. Therefore, this DAP system is relatively dark, as compared to $C_B + O_N V_B$, which might lead to a low ODMR signal. Moreover, we also need to point it out that the ground

state of this DAP is $S = 1$ while it is $S = 1/2$ in $C_B + O_N V_B$, so the optical loop that is discussed in Figure 4 in the main text, would be different.

Figure 11. The $C_B + V_B$ DAP system and the electronic structure at ground state. (a) The configurations we considered. (b) The energy levels at ground state with PBE functional. (c) The energy levels at ground state with HSE functional.

Another example is $C_B + C_N V_B$, which is similar to $C_B + O_N V_B$. The ground state is also $S = 1/2$. However, we find the presence of C_B could influence the geometry of $C_N V_B$. The difference is that carbon atom would bond with a nearby nitrogen to have a Stone-Wales-like geometry as we previously investigated [see *Front. Quantum Sci. Technol.*, 1, 1007756 (2022)]. When the two defects are close to each other, the reconstruction could happen while the long separation between them makes the $C_N V_B$ retrieve the configuration like $O_N V_B$. Unlike the oxygen, the carbon atom still has one dangling bond so there are more localized states in gap. We could still identify the localized states from C_B since they are empty and there is no energy splitting between spin-up and down channels. It shifts upward and then shifts downward with the reconstruction, as labelled in Figure 12b. Similarly, as the distance increases, the empty states from C_B and $C_N V_B$ can somehow converge to a certain point, while the charge transfer could influence the optical excitation. What we proposed in Figure 4 in main text still holds. We believe that these two additional models could validate our explanation.

In addition, we would like to mention $C_B + C_N$ DAP which we have previously investigated. In that case, there is no such kind of local excitation since just single localized state from each defect. So, the optical excitation is only associated with the charge transfer process, while the ground state of

this system is $S = 0$ [Phys. Rev. B 104, 075410 (2021)].

Figure 12. The $C_B + C_N V_B$ DAP system and the electronic structure at ground state. (a) The reconstruction when the two defects are close to each other. (b) The energy levels at ground state with PBE functional. (c) The energy levels at ground state with HSE functional.

4. Please make comparison or justification why this method of identifying defects in hBN is more beneficial or yields more advantages than other methods, say using ZFS or EPR spectra, especially since the proposed method still calculates ZPL which is known to be affected by many parameters, including local strain and lattice vibration.

We cannot say this method is more beneficial or yields more advantages than other methods. Actually, at this point we agree with the Reviewer that the identification of defects in semiconductor requires multiple methods or techniques. With more result from experiment, collected from different methods, there is a higher chance that we can theoretically identify the exact defect structure. ZFS can be used to identify the defect with high spin states ($S \geq 1$) while defects that are singlet or doublet do not have ZFS. EPR is a useful tool to identify paramagnetic defects with unpaired electrons, but it is not applicable for nonmagnetic defect. In addition, the EPR spectrum are complicated in hBN since both boron and nitrogen have almost 100% nonzero nuclear spin and sometimes different defects may have similar EPR spectrum. Also both the ZFS and EPR spectra can be influenced by local environment like strain, etc. Therefore, we have to say that there is no single method that is significantly superior to others in identifying defect in hBN. We added one more section to discuss the defect identification in main text.

5. In identifying a defect in hBN with the DAP model, the authors relied on ADF-STEM results where they were able to image single and multiple carbon and oxygen atoms adjacent to VB monovacancy. What is the typical spatial resolution by this method? Can it identify defects hosted by hBN in deeper regions, rather than the top layer of hBN? The location of a defect could affect the optical signals obtained.

This is an excellent question. The resolution of ADF-STEM with aberration-correction could be up to 1-5 Å [see Figure 4e in Commun. Mater. 4, 19 (2023)]. This enables to see clearly the vacancies on the top layer of hBN. As the reviewer mentioned, the ADF-STEM cannot identify defects hosted in deeper regions, especially when the defect is buried inside the bulk hBN. The optical signal could be different for defects on the surface and inside the sample, too, due to a different dielectric environment. Since a typical resolution of confocal microscopy is hundreds of nanometers, it is sometimes doubtful to say that the optically measured defect is the same as the one observed in ADF-STEM. In this case, the experimental result was achieved on multilayer and monolayer hBN samples. This ensures that the defect is in a top layer, which we believe is a key prerequisite to characterize the defect structure and relate it to the optical signal.

6. In a DAP structure, charge transfer decays rapidly as the distance between the donor and acceptor increases. How do the authors select a cut-off distance?

Indeed, it would be useful to include a larger model to simulate the charge transfer decay between the donor and acceptor with various distance. However, to fully relax the geometries and calculate the electronic structure with the HSE functional, the acceptable size of supercell usually contains hundreds of atoms, for example, 512 atoms in diamond, 256 atoms in hBN in our manuscript. So the possible distance we can consider here is limited by the size of the supercell, which is less than 10 Å. This can be regarded as a cut-off distance in our DFT calculation and we believe the charge transfer could happen easily within this range.

Based on our DFT calculations, we can extract the key parameters that are responsible for the charge transfer and use the DAP model proposed in our manuscript to extrapolate the charge transfer decay to a larger distance, e. g. 20 Å, as shown in Supplementary Figure 5. To calculate the DAP model with distance around 20 Å, a 18*18 supercell with 1024 atoms should be used, which is hard to achieve using the HSE functional.

7. It is not obvious (in the supplementary material) that the presences of oxygen results in a larger boron vacancy. Please provide stronger evidence, preferable quantitatively.

Thank you for the constructive comment. If we understand it correctly, the reviewer would like to see oxygen-related defect in multi-vacancies. As the following figure shows, we simply modelled three larger vacancies with the O_N defect. It is clearly to see the neighbouring boron atoms would bound with each other while the nitrogen atoms do not. For the defect 1 configuration, the paired boron dimer introduces one out-of-plane orbital in gap while the others come from the nitrogen atom. In defect 2, there are only two localized states from two sets of paired boron dimer. In defect 3, besides the paired boron dimer, there is localized state from unpaired single boron atom. And the

rest are from two unsaturated nitrogen atoms similar to $O_N V_B$. In this case, the possible optical excitation is also similar to $O_N V_B$ defect and the calculated ZPL is around 2.5 eV. We believe the multi-vacancies system without oxygen or carbon doping worth a further study. However, a comprehensive investigation will include tens of possible configurations regarding the size of multi-vacancies which is beyond the current scope.

Figure 13. The multi-vacancies system with O_N . The states labelled with orange are from neighboring boron dimer. The states labelled with blue are from single boron dangling bond. The rest are from nitrogen dangling bond.

8. The authors used HSE with mixing parameter $\alpha=0.32$. As the values of ZPL could depend on the value of α , please justify the chosen value. Had the authors optimized the value of α ?

This is a noteworthy point in semiconductor calculation with hybrid functional. The standard HSE06 functional sometimes cannot provide accurate band gaps that are observed in experiment. To reproduce the relative position of the band edges, the mixing parameter α and the screening parameter μ are tuned to ensure the proper piecewise linear behaviour of the total energy as a function of the occupation numbers. We tested the mixing parameter and used $\alpha = 0.32$ to reproduce the experimental optical gap ~ 6.08 eV [Nat. Mater. 3, 404 (2004); Nat. Photonics 10, 262–266 (2016), Phys. Rev. B 51, 6868 (1995)]. This setting has been widely used in current studies, see [ACS Photonics 5, 1967–1976 (2018); arXiv:2403.00755v1; Phys. Rev. B 105, 184101 (2022)]. There are other articles using $\alpha = 0.31$ which could yield a similar band gap of hBN [Phys. Rev. B 97, 214104 (2018); Phys. Rev. Mater. 6, 014005 (2022); ACS Appl. Nano Mater. 7, 16, 18979–18985 (2024)]. We also note $\alpha = 0.40$ is used to have the band gap of 6.42 eV of hBN with zero-point renormalization included [Appl. Phys. Lett. 115, 212101 (2019)]. This might overestimate the optical transition energy of defects and hBN.

We here plot the Kohn-Sham levels of $O_N V_B^-$ with different mixing parameter. The band gap

increases from 5.9 to 6.4 eV, but the energy difference between the localised levels increases by only ~ 0.1 eV, indicating that the small variations of mixing parameter do not significantly influence the optical properties of defect at this level.

Figure 14. The influence of mixing parameter on the ZPL of $O_N V_B^-$.

9. When performing geometrical optimization, do the authors allow only the atoms in the hBN plane to relax, or do they allow all atoms in the supercell to relax? Otherwise, please argue why different ways of geometrical optimization would not affect the desired optical properties?

During optimization, all the atoms are fully relaxed. Based on our previous studies and experience, we have to say the neighbouring layers would not change the optical properties of in-plane defects since the localized wavefunction is confined in hBN plane. However, the interaction with neighbouring layers should be included and carefully considered for out-of-plane defects [ACS Appl. Mater. Interfaces 13,45768–45777 (2021)]. At this condition, two-layer supercell model is not enough to avoid the defect-defect interaction in the z direction.

10. The energy cut-off seems a bit low. Could the authors include the convergence or optimization tests in the supplementary material? What is the energy convergence threshold? How is the charge correlation computed?

The cutoff energy has been tested in our previous studies and this does not significantly change the optical properties. We here add the cutoff convergence test with PBE functional on primitive cell of hBN. The energy difference from 450 eV to 600 eV is within 5 meV. We choose 450 eV to accelerate the computation of excited states.

Figure 15. Test on cutoff energy with hBN primitive cell by PBE functional.

11. I would request that the authors add a paragraph on different methods for identifying qubit defects hosted by hBN, their advantages and disadvantages. This should give more transparent motivation and benefits of this work.

Thank you for the constructive suggestion. Generally, the atomic structure of optical emitters in hBN is challenging to determine experimentally. Direct imaging through the high-resolution transmission electron microscopy can provide geometric information about defects on the top layer of hBN. However, the optical signals sometimes originate from defects located deeper within the crystal. EPR is a powerful tool for identifying paramagnetic point defects (with one or more unpaired electrons), based on their zero-field splitting (ZFS) and hyperfine interactions. In hBN, the interpretation of EPR signals is challenging but feasible in principle, because both boron and nitrogen have 100% natural abundance of non-zero nuclear spin isotopes—a factor that increases the complexity of extracting the appropriate spin Hamiltonian. Furthermore, the ZFS of high-spin defects provides an important parameter for theoretical benchmarking. However, EPR cannot detect non-magnetic defects, and the observed signals may originate from multiple overlapping defect types.

Confocal microscopy and photon antibunching measurements can offer optical insights, such as ZPL energy, phonon sideband (PSB), and fluorescence lifetime of individual defects. Still, distinct defects may produce similar optical features, leading to potential misassignments. By applying external field—such as strain or electric field—the optical response of the emitters can be modulated, offering additional information for defect identification. In our model, we propose that the optical signal could originate from spin pairs rather than isolated single defects. Multiple optical transition pathways within such spin-pair systems can lead to spin polarization and ODMR contrast, providing another route for identifying quantum defects.

In summary, a comprehensive analysis of SPEs or qubits in hBN requires a multi-dimensional approach that integrates structural, magnetic, optical, and field-dependent characterizations.

12. More details about numerical simulations of Eq.(3) should be provided in the methodology section? For instances, what numerical method was employed? How the parameter values (e.g. D-tensor) were chosen?

Eq. 3 could be solved with Easyspin code with input parameters we get from DFT calculation. The equation contains zero field interaction, hyperfine interaction and electron Zeeman interaction. Of course, the nuclear Zeeman interaction and quadrupole interaction could also be considered but usually they are small. The zero field splitting D tensor was calculated using VASP as implemented by Martijn Marsman [J. Phys. Condens. Matter 26, 015305 (2014)]. A further correction scheme was applied due to the spin contamination [Phys. Rev. Res. 2, 022024 (2020)]. The hyperfine tensor A was also calculated through DFT with a Fermi contact term included [Phys. Rev. B 77, 155206 (2008)]. The spin exchange energy J cannot be calculated accurately with DFT method; however, it should be small so we assume J in the MHz region. With these parameters, it is possible to simulate the spin Hamiltonian under an external magnetic field.

Miscellaneous

1. In all figures, please include the labels for each type of atoms. The authors have done so in the caption, but I think it is helpful to do so in each figure as well.

The figures have been updated according to the suggestion.

2. Also, in the first simplified electronic structures in the main text, the authors should explain the symbols (e.g., a_1 , b_1 , b_2 , filled and unfilled triangles), rather than referring to the supplementary material. This should make the article more translucent and self-contained, and should help the readers more.

Symbols like a_1 , b_1 , b_2 are the irreducible representations of localized wavefunctions of defects. The filled and unfilled triangles denote the empty and occupied defect states in gap. We updated the figure captions according to the Reviewer's suggestion.

3. What does IS stand for in Fig. 4a?

The IS (intermediate states) should be changed to MS which means the metastable states. We changed the figure accordingly to be consistent with the description in main text.

4. In the supplementary material, Fig. 5 (below Eq.(1)) should be written as "Supplementary Figure 5" to keep to notation consistent. There are only four figures in the main text.

Thank you for pointing this out. We modified the supplementary information accordingly.

5. The article's title appears to be too broad and generalized, hence over-claims the presented results. Until there are more DAP structures to validate these results, I think it is more appropriate to specify systems in the title.

We respectfully disagree. In our original manuscript we already considered two different DAP models. In the revised manuscript we added further DAP coupled spin systems to the discussion in the Supplementary Information, thus we think that the title is appropriate.

Reviewer #4 (Remarks to the Author):

We also thank you for your constructive comments for our manuscript.

ANSWER TO REFEREES

We thank all the Referees for their positive comments. We further revised our manuscript according to the comments from reviewers.

Reviewer #1 (Remarks to the Author):

I maintain the opinion I expressed in my first review that the presented results are robust, interesting, and important. Moreover, after revision, they are now presented in a much clearer and more accessible manner. Honestly, I found the revised manuscript much more enjoyable to read compared to the initial version.

I recommend the manuscript for publication.

The only remark I would like to make is that the formation energy of CB-VBON and ON-VBON depends on the chemical potential during growth (e.g., N-poor or N-rich conditions). Therefore, it should be clarified to which specific conditions the presented values correspond.

R: The initial formation energy was at N-poor condition. We further plot the formation energy of the three DAPs under N-rich condition. We modify the description in SM accordingly.

Reviewer #2 (Remarks to the Author):

I thanks the author for the response.

I do not have additional questions, the authors addressed all my concerns and replied to my questions, so I recommend the paper for publication.

Reviewer #3 (Remarks to the Author):

The authors have fairly answered my questions. I recommend the manuscript to be published in Nature Communication after these minor revisions.

1. Eq.(2) should be checked on the subscripts. I would also request that each Hamiltonian be stated explicitly.

R: We correct the subscripts in Eq. 2. The description of each Hamiltonian is mentioned at two paragraphs before the Eq. 2 in main text.

2. The dielectric constant value should be stated in the manuscript.

R: We add the dielectric constant in main text.

3. There are several awkward sentences or phrases in the Supplementary (e.g. first line below Section V, also a few lines after that).

R: We further polished the entire Supplementary Materials.

4. Please provide citation that IC is significantly faster than ISC.

R: We add one reference in the main text.

5. I would request that the abstract be revised and added more information about the results to reflect the revised manuscript. In the current version, only one sentence in the abstract is about the results.

R: We have two sentences about the results in the abstract of the main text. Because of the word-limit of the abstract we extended the second sentence about the microscopic mechanism (marked by red ink):

Here we connect the variance in the optical spectra, optical lifetimes and spectral stability of quantum emitters to donor-acceptor pairs (DAP) in hBN by means of ab initio calculations. We find that DAPs can exhibit ODMR signal for the acceptor counterpart of the defect pair with $S = 1/2$ ground state at non-zero magnetic fields, depending on the donor partner and dominantly mediated by the hyperfine interaction.

The paper “Quantum Emission from Coupled Spin Pairs in Hexagonal Boron Nitride” by Li et al investigate the donor-acceptor pair composed by the primary acceptor defect $O_N V_B$ and surrounding donor O_N , C_B . The authors discuss the energy diagram related to various transitions and charge states as a function of donor-acceptor separation. The ODMR mechanism is qualitatively explored in relation to energy diagrams and relative recombination rates. The authors suggest that the spin mixing of two metastable states could explain the observed spin polarization and ODMR contrast, with transition rates being sensitive to donor-acceptor separation and potential engineering. Below are a few points I found unclear and believe could benefit from clarification:

1. The work considers the charged donor-acceptor pair and the neutral donor-acceptor pair as two distinct states. For instance, in the statement, “Furthermore, the total energy of $CB^+ - ONVB^-$ configuration is lower than that of $CB^0 - ONVB^0$ ”, the energy difference between the two configurations is compared. It remains unclear how the total energy is calculated for systems with and without local charge transfer. Is it determined by DFT total energy computations for different electronic configurations based on the atomic model shown in Figure 1a? Additionally, is this claim valid for all 10 configurations proposed in the main text?
2. The bright ZPL emissions are attributed to the a_1 to b_1 transition from the $(O_N V_B)^{-1}$ site. The paper also mentions that the charge transfer process from $(O_N V_B)^{-1}$ to C_B^+ can also luminesce, resulting in two metastable states of $S=1/2$ and $S=3/2$. Supplementary Figure 3 depicts the luminescence energy for the ground state to the $S=3/2$ metastable state and compares it to the ZPL energy of the internal a_1 to b_1 transition (main text Figure 1b). However, there are two points of confusion:
 - a. The supplementary figure denotes energy as the “*energy difference of Kohn-Sham levels*,” which typically refers to the single-particle energy level differences. Meanwhile, the main text states, “*Supplementary Figure 3 shows the Kohn-Sham energy difference*,” suggesting the total energy difference between the ground state and excited state configurations. If it indeed refers to the total energy difference, why is there such a significant difference between the bright transition energy in Supplementary Figure 3 and the ZPL energy in Figure 1b?
 - b. The distinction between the ZPL associated with charge-transfer-related transitions and the internal $a_1 \rightarrow b_1$ transition at $(O_N V_B)^{-1}$ is unclear. If both are derived from constrained DFT with different single-particle level configurations of the same DAP model, it would be helpful for the authors to clarify the transitions or configurations in the single-particle diagram.

Including an electronic configuration diagram or group theory analysis would enhance understanding and distinguishment of the metastable $S=1/2$ and $S=3/2$ states, as well as the ground, excited, the dark transition, dim transition in Supplementary Fig3.

3. Equation (1) Model and Energy Interpretation:

Equation (1) aims to explain the emission mechanism related to charge-transfer luminescence. However, it lacks sufficient explanation of its physical meaning, particularly regarding the definition of $E(R_i)$ represents the emission energy, the substantial energy differences between Supplementary Figures 3 and 4 (dark transitions) require further discussion and explanation. For example, in the statement, "The total energy of $S = 3/2$ state lies with 1.68 eV to 2.36 eV higher than that of the ground state depending on the DAP distance and this means that the bright excited state (~ 2.0 eV) and the metastable states can switch their energy position." the origin of these energy values is unclear.

4. Consistency Between Figures and Models:

In the discussion, Figures 3b and 3c present a donor-acceptor model. It would be beneficial to explicitly connect these states to the energy diagram in Figure 4a for improved clarity.

5. Radiative and Non-Radiative Processes:

The transition r_2 and r_3 account for both radiative and non-radiative processes. However, their discussion appears loosely organized. For instance, the radiative transition dipole moment is computed in the Results section, while non-radiative lifetimes (internal conversion rates) are only discussed towards the end of the Discussion section. Since energy level differences influence transition rates, corresponding data could be more clearly presented.

Specific details on how internal conversion rates were calculated should be provided, along with citations. A table summarizing the energy levels of the ground state (GS), excited state (ES), and metastable states (Ds and Qs) would be very helpful. Additionally, the method for extracting electron-phonon coupling requires further explanation.

6. The following conclusion appears inconsistent: "the small energy difference between the optically active doublet excited state and the dim metastable state and the strong electron-phonon coupling lead to fast internal conversion rate around 10^{10} MHz. Therefore the system relaxes back to ground state rapidly without detectable optical signal." While the first part asserts that the transition

Reviewer #2 (Remarks to the Author):

from the excited state (ES) to the dim state (Ds) is rapid (r_1), the subsequent statement focuses on the fast and fully non-radiative relaxation from Ds to the ground state (r_2, r_3). The reasoning appears inconsistent and could be clarified.

For these reason we cannot recommend publication right now and the manuscript requires revision

Reviewer's Report: NCOMMS-24-57066

The authors related and explained the variance in the optical spectral, optical lifetimes and spectral stability of quantum emitters (QE) to donor-acceptor pairs (DAP) in hexagonal boron nitride (hBN). They found that DAPs can exhibit optically detected magnetic resonance (ODMR) signals for the acceptor counterpart of the defect pair with $S = 1/2$ ground state at non-zero magnetic fields depending on the donor partner, and claimed that the DAP model and its transition mechanisms provide a recipe towards defect qubit identification and performance optimization in hBN.

General Comments

Identifying a microscopic origin of single photons or optically addressable qubits, especially ones that operator at room temperature, is immensely important and has far-reaching impact in quantum information processing. Many techniques have been employed, from which we can obtain different physical quantities. However, it is not clear which property is a smoking gun to identify a qubit origin. This research topic is very active and of high interest to the quantum technology community.

The article is well-organized, well-written, comprehensible and easy to follow and contributes significantly to a development in the field. More importantly, it fills in the literature gap for explaining the 2-eV emission from first-principles, rather than phenomenologically. Given the relevance and timely research of the topic, I would recommend the article be published for the community. However, in my opinion, the article is still lacking the novelty, and it should include more DAP systems to validate the claimed results. On the novelty argument, DAP has been reported before, and well documented, as pointed out in many references by the authors (e.g. Phys. Rev. B (2021), **104**, 075410) and also Nano Lett. (2022), **22**, 1331-1337. Therefore, I would recommend that the article be revised, taking the following comments into account. More essentially, the authors should explain why other mechanisms of photon emission at around 2 eV must be ruled out, and include more results from other DAP systems.

Comments on Content

1. The authors claimed that the variance in the optical properties comes from the type and location of the donor partner. How can they rule out effects of strain, or other mechanisms?
2. It is well known that optical signals alone can provide insufficiently information on the chemical nature of a defect, as local environment (e.g. strain, isotopes, charges) can affect the PL spectra. The authors should clearly explain why their method can uniquely determine a microscopic origin of a qubit in hBN.
3. Since the authors claimed that DAP emission is responsible for most of the visible ZPL emission at around 2.0 eV, and DAPs are quite common structures, they should validate their results with more systems other than the $O_N-O_NV_B$ and $C_B-O_NV_B$ systems. Could the authors include results from more DAP systems to validate their model? Maybe in the supplementary material.
4. Please make comparison or justification why this method of identifying defects in hBN is more beneficial or yields more advantages than other methods, say using ZFS or EPR spectra, especially since the proposed method still calculates ZPL which is known to be affected by many parameters, including local strain and lattice vibration.

5. In identifying a defect in hBN with the DAP model, the authors relied on ADF-STEM results where they were able to image single and multiple carbon and oxygen atoms adjacent to V_B monovacancy. What is the typical spatial resolution by this method? Can it identify defects hosted by hBN in deeper regions, rather than the top layer of hBN? The location of a defect could affect the optical signals obtained.
6. In a DAP structure, charge transfer decays rapidly as the distance between the donor and acceptor increases. How do the authors select a cut-off distance?
7. It is not obvious (in the supplementary material) that the presences of oxygen results in a larger boron vacancy. Please provide stronger evidence, preferable quantitatively.
8. The authors used HSE with mixing parameter $\alpha = 0.32$. As the values of ZPL could depend on the value of α , please justify the chosen value. Had the authors optimized the value of α ?
9. When performing geometrical optimization, do the authors allow only the atoms in the hBN plane to relax, or do they allow all atoms in the supercell to relax? Otherwise, please argue why different ways of geometrical optimization would not affect the desired optical properties?
10. The energy cut-off seems a bit low. Could the authors include the convergence or optimization tests in the supplementary material? What is the energy convergence threshold? How is the charge correlation computed?
11. I would request that the authors add a paragraph on different methods for identifying qubit defects hosted by hBN, their advantages and disadvantages. This should give more transparent motivation and benefits of this work.
12. More details about numerical simulations of Eq.(3) should be provided in the methodology section? For instances, what numerical method was employed? How the parameter values (e.g. D-tensor) were chosen?

Miscellaneous

1. In all figures, please include the labels for each type of atoms. The authors have done so in the caption, but I think it is helpful to do so in each figure as well.
2. Also, in the first simplified electronic structures in the main text, the authors should explain the symbols (e.g., a_1 , b_1 , b_2 , filled and unfilled triangles), rather than referring to the supplementary material. This should make the article more translucent and self-contained, and should help the readers more.
3. What does IS stand for in Fig. 4a?
4. In the supplementary material, Fig. 5 (below Eq.(1)) should be written as "Supplementary Figure 5" to keep to notation consistent. There are only four figures in the main text.
5. The article's title appears to be too broad and generalized, hence over-claims the presented results. Until there are more DAP structures to validate these results, I think it is more appropriate to specify systems in the title.